# Validation of a deep learning, value-based care model to predict mortality and comorbidities from chest radiographs in COVID-19

**Ayis Pyrros**[1][☯]*, **Jorge Rodriguez Fernandez**[2][☯], **Stephen M. Borstelmann**[3], **Adam Flanders**[4], **Daniel Wenzke**[5], **Eric Hart**[6], **Jeanne M. Horowitz**[6], **Paul Nikolaidis**[6], **Melinda Willis**[1], **Andrew Chen**[7], **Patrick Cole**[7], **Nasir Siddiqui**[1], **Momin Muzaffar**[1], **Nadir Muzaffar**[1], **Jennifer McVean**[8], **Martha Menchaca**[9], **Aggelos K. Katsaggelos**[10], **Sanmi Koyejo**[7], **William Galanter**[11]

1 Department of Radiology, Duly Health and Care, Hinsdale, Illinois, 2 Department of Neurology, University of Illinois at Chicago, Chicago, Illinois, 3 Department of Radiology, University of Central Florida, Orlando, Florida, 4 Department of Radiology, Thomas Jefferson University Hospital, Philadelphia, Pennsylvania, 5 Department of Radiology, NorthShore University HealthSystem, Evanston, Illinois, 6 Department of Radiology, Northwestern University, Chicago, Illinois, 7 Department of Computer Science, University of Illinois at Urbana- Champaign, Urbana-Champaign, Illinois, 8 Medtronic, Minneapolis, Minnesota, 9 Department of Radiology, University of Illinois at Chicago, Chicago, Illinois, 10 Department of Electrical and Computer Engineering, Northwestern University, Evanston, Illinois, 11 Department of Medicine, University of Illinois at Chicago, Chicago, Illinois

☯ These authors contributed equally to this work.
* ayis@ayis.org

**Data Availability Statement:** Source code available: https://zenodo.org/record/6587719#.YpD1zC-cZQI.

## Abstract

We validate a deep learning model predicting comorbidities from frontal chest radiographs (CXRs) in patients with coronavirus disease 2019 (COVID-19) and compare the model's performance with hierarchical condition category (HCC) and mortality outcomes in COVID-19. The model was trained and tested on 14,121 ambulatory frontal CXRs from 2010 to 2019 at a single institution, modeling select comorbidities using the value-based Medicare Advantage HCC Risk Adjustment Model. Sex, age, HCC codes, and risk adjustment factor (RAF) score were used. The model was validated on frontal CXRs from 413 ambulatory patients with COVID-19 (internal cohort) and on initial frontal CXRs from 487 COVID-19 hospitalized patients (external cohort). The discriminatory ability of the model was assessed using receiver operating characteristic (ROC) curves compared to the HCC data from electronic health records, and predicted age and RAF score were compared using correlation coefficient and absolute mean error. The model predictions were used as covariables in logistic regression models to evaluate the prediction of mortality in the external cohort. Predicted comorbidities from frontal CXRs, including diabetes with chronic complications, obesity, congestive heart failure, arrhythmias, vascular disease, and chronic obstructive pulmonary disease, had a total area under ROC curve (AUC) of 0.85 (95% CI: 0.85–0.86). The ROC AUC of predicted mortality for the model was 0.84 (95% CI,0.79–0.88) for the combined cohorts. This model using only frontal CXRs predicted select comorbidities and

**Funding:** AP, NS and SK were funded by the Medical Imaging Data Resource Center, which is supported by the National Institute of Biomedical Imaging and Bioengineering of the National Institutes of Health under contracts 75N92020C00008 and 75N92020C00021. JR-F and WG received funding from the University of Illinois at Chicago Center for Clinical and Translational Science (CCTS) award ULTR002003. The funders had no role in study design, data collection and analysis, decision to publish, or preparation of the manuscript.

**Competing interests:** The authors have declared that no competing interests exist.

RAF score in both internal ambulatory and external hospitalized COVID-19 cohorts and was discriminatory of mortality, supporting its potential use in clinical decision making.

## Author summary

Artificial Intelligence algorithms in Radiology can be used not only on standard imaging data like chest radiographs to predict diagnoses but can also incorporate other data. We wanted to find out if we could combine administrative and demographic data with chest radiographs to predict common comorbidities and mortality. Our deep learning algorithm was able to predict diabetes with chronic complications, obesity, congestive heart failure, arrythmias, vascular disease, and chronic obstructive pulmonary disease. The deep learning algorithm was also able to predict an administrative metric (RAF score) used in value-based Medicare Advantage plans. We used these predictions as biomarkers to predict mortality with a second statistical model using logistic regression in COVID-19 patients both in and out of the hospital. The degree of discrimination both the deep learning algorithm and statistical model provide would be considered 'good' by most, and certainly much better than chance alone. It was measured at 0.85 (95% CI: 0.85–0.86) by the area under the ROC curve method for the artificial intelligence algorithm, and 0.84 (95% CI:0.79–0.88) by the same method for the statistical mortality prediction model.

## Introduction

Managed care, also known as value-based care (VBC) in the US, emphasizes improved outcomes and decreased costs by managing chronic comorbidities and grouping (ICD10) diagnosis codes with reimbursements proportioned to disease burden [1]. There is growing concern that these VBC models do not recognize radiology's central role [1]. Radiologists are often unfamiliar with the complexity of VBC systems and their significance to clinical practice [2]. Furthermore, radiologists frequently receive limited relevant clinical information on the radiology request; instead, clinical information is buried in the non-radiology electronic health record (EHR) data which is at best awkward and time-consuming to retrieve and at worst simply un-obtainable.

The risk adjustment factor (RAF) score, also called the Medicare risk adjustment, represents the amalgamation of the ICD10 hierarchical condition categories (HCCs) for a patient [3]. The 2019 version 23 model includes 83 HCCs, comprising over 9,500 ICD10 codes, each having a numeric coefficient. There are additional coefficients for HCC interactions (disease interactions) and demographics composed of age and sex for patients over the age of 65 years (S1 Table). The RAF score directly correlates with disease burden, and likewise future healthcare costs, with the mean value calibrated to 1.0. A score of 1.1 indicates an approximately 10% higher reimbursement for a patient as the predicted costs are higher. The RAF score is calibrated yearly through updates to the Centers for Medicare and Medicaid Services (CMS) model coefficients and HCCs. In this paper we used model version 23 from 2019. The HCC codes are generated through encounters with healthcare providers and recorded in administrative data. As such, these data elements are often more reproducible and amenable to analysis than manual EHR review. These administrative data have been shown to predict mortality in patients with coronavirus disease 2019 (COVID-19) [4].

The COVID-19 pandemic has strained healthcare systems worldwide, amplifying existing deficiencies. While most infected individuals experience mild or no symptoms, some become

severely ill, require prolonged hospitalization [5], and succumb to the disease. Comorbidities such as diabetes, cardiovascular disease, and morbid obesity are associated with worse outcomes [6–8]. Currently, comorbidity data are extracted from contemporaneously provided patient history, manual records, and/or EHRs [9]. Those methods are imperfect, often incomplete, difficult to replicate across institutions, or not available for physician decision-making.

Multiple predictive clinical models of the course of COVID-19 infection have been developed utilizing demographic information, clinically obtained data regarding comorbidities, laboratory markers, and radiography [10,11]. In these models, radiography has been incorporated by quantifying the geographic extent and degree of lung opacity [10,12]. Using a convolutional neural network (CNN) to "link" HCCs and RAF scores to radiographs can convert the images into useful biomarkers of patients' chronic disease burden. Frontal chest radiographs (CXRs) have been used to directly predict or quantify patient comorbidities that contribute to outcomes [13].

We hypothesize that VBC data and CXRs can be used to train a deep learning (DL) model to predict select comorbidities and RAF scores and that these predictions can be used to model COVID-19 mortality. The purpose of this study was to validate a DL algorithm that could predict relevant comorbidities from CXRs, trained on administrative data from a VBC model, and validate this methodology in a distinct external blinded cohort of COVID-19 patients.

## Methods

This retrospective study was approved on scientific and ethical research basis by the institutional review boards of both institutions and was granted waivers of Health Insurance Portability and Accountability Act authorization and written informed consent. This research study was conducted retrospectively from data obtained for clinical purposes.

### Image selection, acquisition and pre-processing

CXRs for the base training set developed from 14,121 posteroanterior (PA) CXRs done from 2010 to 2019 at Duly Health and Care, a large suburban multi-specialty practice, were obtained conventionally with digital PA radiography. CXRs from the external validation set were portable anteroposterior (AP) radiographs, except for 30 conventionally obtained AP radiographs. All CXRs were extracted from a picture archiving and communication system (PACS) system utilizing a scripted method (SikuliX, 2.0.2) and saved as de-identified 8-bit grayscale portable network graphics (PNG) files for the training and internal validation sets and 24-bit Joint Photographic Experts Group (JPEG) files for the external validation set. Radiographs from the external validation set had white lettering embedded, stating "PORTABLE" and "UPRIGHT" on the top corners of the images. Images were resized to 256 × 256 pixels and base weights generated by training the model with an 80%/20% train/validation split.

To lessen the possibility of inadvertently creating a "PORTABLE/UPRIGHT" PA/AP radiograph detector, or PNG/JPG detector as a confounder, pre-processing of the images with sanity checks were utilized. As the larger embedded white lettering was not present in the training data nor the internal validation set, similar embedding on the internal validation set was digitally augmented and tested. Additionally, occlusion mapping techniques were utilized, described subsequently. Additional testing of PNG and JPEG file formats confirmed that the DL model was not affected by the initial file format prior to conversion.

The internal validation set (internal COVID+ test set) (N = 413) was seen between 3/17/2020 and 10/24/20 and received both a CXR and positive real-time reverse transcription polymerase chain reaction (RT-PCR) COVID-19 test in the ambulatory or immediate care setting. Some of the patients went to the emergency department after the positive RT-PCR test, and

some were hospitalized. The EHR clinical notes were reviewed to determine the reason and date of admission if any occurred.

The external validation set (external COVID+ test set) (N = 487) was seen at a large urban tertiary academic hospital University of Illinois Chicago between 3/14/2020 and 8/12/2020 and received a frontal CXR in the emergency department and a positive RT-PCR COVID-19 test.

## Clinical data and inclusion criteria

Clinical variables included patient sex, age, mortality, morbidity, history of chronic obstructive pulmonary disease (COPD), diabetes with chronic complications, morbid obesity (body mass index [BMI] > 40), congestive heart failure (CHF), cardiac arrhythmias, and vascular disease as determined by ICD10 codes. These common comorbidities were chosen as they are linked with HCC codes. The following HCCs were used: diabetes with chronic complications (HCC18), morbid obesity (HCC22), CHF (HCC85), specified heart arrhythmias (HCC96), vascular disease (HCC108), COPD (HCC111). Complete RAF scores, excluding age and sex, with the RAF score coefficients from the v23 model community, nondual-aged were used [14]. Administrative RAF scores were calculated from ICD10 codes using Excel (Excel 16; Microsoft, Redmond, Wash), excluding the demographic components. The calculated RAF score included all the available HCC coefficients for the patient, not just the six aforementioned HCC coefficients.

In cases of multiple RT-PCR COVID-19 tests, or negative and then positive tests, the first positive test was used as the reference date. In patients with multiple CXRs, only the radiograph closest to the first positive RT-PCR test was used (one radiograph per subject). Patients without locally available or recent CXRs, those with radiographs obtained more than 14 days from positive RT-PCR testing, and subjects less than 16 years of age at the time of radiography were excluded. Mortality was defined by death prior to discharge (Fig 1). The outcome was determined by chart review in both cohorts.

## Deep learning

We created a multi-task multi-class CNN classifier by adapting a baseline standard ResNet34 classification model. We modified the basic ResNet building block by changing the 3x3 2D Convolution layer in the block to a 2D CoordConv layer to benefit from the additional spatial encoding ability of CoordConv [15]. CoordConv allows the convolution layer access to its own input coordinates, using an extra coordinate channel. Using standard train/test/validation splits to isolate test data from validation data, the CoordConv ResNet34 model was first trained on publicly available CXR data from the CheXpert dataset [16] and was then fine-tuned on local data using the internal anonymized outpatient frontal CXRs from 2010 to 2019.

Technical and hyperparameter details are as follows. The training was performed on a Linux (Ubuntu 18.04; Canonical, London, England) machine with two Nvidia TITAN GPUs (Nvidia Corporation, Santa Clara, Calif), with CUDA 11.0 (Nvidia) for 50 epochs over 10.38 hours. Training used image and batch sizes of 256 × 256 pixels and 64, respectively. All programs were run in Python (Python 3.6; Python Software Foundation, Wilmington, Del) and PyTorch (version 1.01; pytorch.org). The CNN was trained by ADAM at a learning rate of 0.0005 with the learning rate decreased by a factor of 10 when the loss ceased to decrease for 10 iterations [17]. Binary cross-entropy was used as the objective function for HCCs and sex classes, mean squared error for age and RAF classes. Data augmentation of images was performed with random horizontal flips (20%), random rotations (+/- 10 degrees), crops (range 1.0, 1.1), zooms (range 0.75, 1.33), random brightness and contrast (range 0.8,1.2), and skews

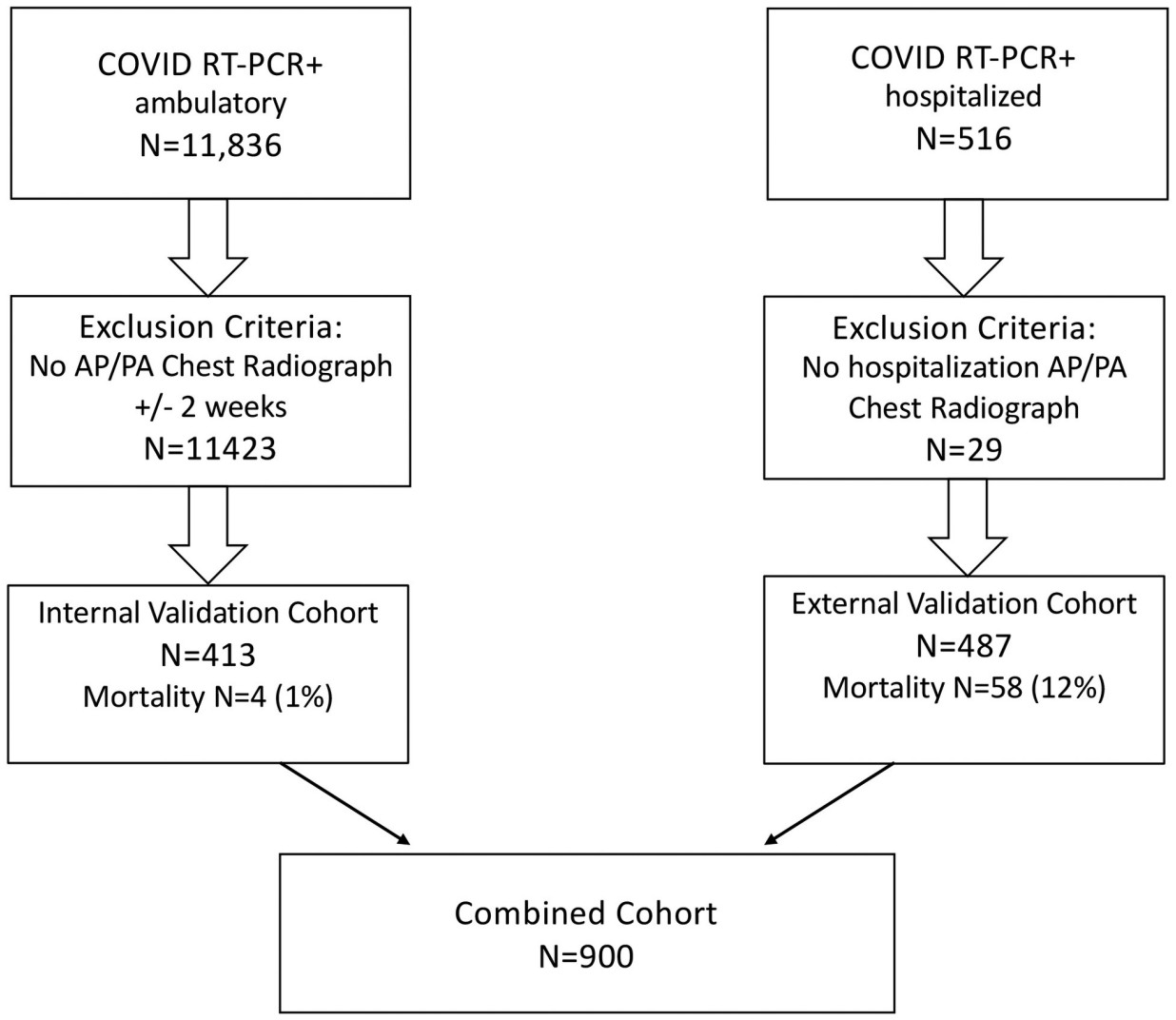

**Fig 1. Flowchart of patient inclusion.** Patients with absent or negative real-time reverse transcription polymerase chain reaction (RT-PCR) test results were excluded. COVID-19 = coronavirus disease 2019.

(distortion scale of 0.2, with a probability 0.75). Images were normalized with the mean and standard deviation of the pixel values computed over the training set. The PIL library was used for image resizing of the initial JPEG and PNG files. We report the hyperparameters for reproducibility purposes.

Positive pixel-based occlusion-based attribution maps were generated, using the Python library Captum 0.3.1, in which areas of the image are occluded and then used to quantify how the model's prediction changes for each class [18]. We used a standard sliding window of size $15 \times 15$ with a stride of 8 in both image dimensions. At each location, the image is occluded with a baseline value of 0. This technique does not alter the DL model, but rather perturbs only the input image [18].

## Statistical analysis

Demographic characteristics and clinical findings were compared between the internal and external COVID+ cohorts using $\chi^2$ for categorical variables and t-tests for continuous variables

**Table 1. Demographics, Clinical Findings and Deep Learning Chest Radiograph Characteristics per COVID-19 Cohort.**

| Characteristic[a] | Internal validation (N = 413) | External validation (N = 487) | Comparative P Value |
|---|---|---|---|
| Age, mean (SD) | 49.9 (16.1) | 56.3 (16.4) | < .001 |
| Sex | | | .251 |
| Male | 222 (53.8%) | 242 (49.7%) | |
| Female | 191 (46.2%) | 245 (50.3%) | |
| Race | | | < .001 |
| White | 222 (53.8%) | 32 (6.6%) | |
| African American | 27 (6.5%) | 225 (46.2%) | |
| Hispanic | 93 (22.5%) | 59 (12.1%) | |
| Other | 46 (11.1%) | 165 (33.9%) | |
| BMI, mean (SD) | 30.8 (7.08) | 32.2 (10.1) | .011 |
| Mortality* | 4 (1.0%) | 58 (11.9%) | < .001 |
| Medical conditions per EHR | | | |
| Diabetes with complications | 44 (10.7%) | 172 (35.8%) | < .001 |
| Morbid obesity | 39 (9.4%) | 112 (23.3%) | < .001 |
| Congestive heart failure | 11 (2.7%) | 104 (21.6%) | < .001 |
| Cardiac arrhythmias | 19 (4.6%) | 71 (14.8%) | < .001 |
| Vascular disease | 30 (7.3%) | 105 (21.8%) | < .001 |
| COPD | 7 (1.7%) | 57 (11.9%) | < .001 |
| RAF, mean (SD) | 0.246 (0.493) | 1.51 (1.66) | < .001 |
| Frontal chest radiograph DL outcomes | | | |
| Predicted age, mean (SD) | 53.2 (13.4) | 60.7 (10.4) | < .001 |
| Medical conditions, mean (SD)[b] | | | |
| Diabetes with complications | 0.182 (0.197) | 0.534 (0.208) | < .001 |
| Morbid obesity | 0.165 (0.250) | 0.353 (0.332) | < .001 |
| Congestive heart failure | 0.108 (0.161) | 0.426 (0.252) | < .001 |
| Cardiac arrhythmias | 0.080 (0.134) | 0.286 (0.221) | < .001 |
| Vascular disease | 0.237 (0.232) | 0.412 (0.212) | < .001 |
| COPD | 0.084 (0.146) | 0.136 (0.147) | < .001 |
| Predicted RAF, mean (SD) | 0.610 (0.463) | 1.451 (0.528) | < .001 |

[a]Data are given as numbers (percentages) for each group, unless otherwise specified.

[b]Mean index is not a percent or likelihood.

*Mortality is different between the two cohorts as the internal cohort was ambulatory while the external cohort was hospitalized.

Abbreviations: BMI = body mass index, COPD = chronic obstructive pulmonary disease, COVID-19 = coronavirus disease 2019, EHR = electronic health record, RAF = risk adjustment factor

(Table 1). The predictions for age, RAF score, and six HCCs were compared to administrative data for the internal and external COVID+ cohorts to test the model's performance in predicting comorbidities. The analysis used AUC ROC. Age and RAF score were compared to the administrative data with correlation coefficients. The Spearman's rank correlation test was used to assess the similarity between the DL predictions and EHR for each medical condition and for RAF score. In Table 2, the eight AUCs were individually compared using the method of DeLong, and due to the use of multiple comparisons, the P values were adjusted using the method of Holm-Bonferroni [19,20]. In Table 3, recall and precision values are demonstrated.

Multivariable logistic regression, backward elimination with fractional polynomial transformation, was performed with $P < .05$ for inclusion of the variables from the DL model: sex, age, and six common ICD10 HCC codes (model v23) [21]. Feature selection was performed by selecting the ten most prevalent conditions and using backward elimination based on area

**Table 2. Multi-Task Deep Learning HCC-Based Comorbidity and Sex Predictions for Internal and External Cohorts.**

| Characteristic | Internal cohort AUC (95% CI) | | External cohort AUC (95% CI) | | *P* value |
|---|---|---|---|---|---|
| Sex | 0.94 | (0.91–0.96) | 0.97 | (0.95–0.98) | < .05 |
| Morbid obesity | 0.91 | (0.88–0.95) | 0.86 | (0.83–0.89) | .055 |
| Congestive heart failure | 0.84 | (0.74–0.93) | 0.71 | (0.65–0.76) | < .05 |
| Vascular disease | 0.87 | (0.82–0.91) | 0.70 | (0.64–0.75) | < .001 |
| Cardiac arrhythmias | 0.76 | (0.66–0.86) | 0.74 | (0.68–0.80) | .752 |
| COPD | 0.85 | (0.71–0.98) | 0.71 | (0.63–0.78) | .075 |
| Diabetes with chronic complications | 0.77 | (0.69–0.84) | 0.63 | (0.58–0.68) | < .005 |
| All conditions | 0.85 | (0.82–0.88) | 0.76 | (0.73–0.78) | < .001 |

Abbreviations: AUC = area under the receiver operating characteristic curve, CI = confidence interval, COPD = chronic obstructive pulmonary disease, HCC = hierarchical condition category.

under the receiver operating characteristic (ROC) curve (AUC) analysis discussed under statistical analysis.

Logistic regression with produced odds ratios (ORs) and 95% confidence intervals (CIs) for the outcomes of mortality on both COVID+ cohorts was performed (Table 4). Model calibration was assessed graphically using nonparametric bootstrapping. The external cohort mortality model was then applied to the combined cohort. All tests were two-sided, $P < .05$ was deemed statistically significant, and analysis was conducted in R version 4 (R Foundation for Statistical Computing, Vienna, Austria).

## Results

### Patient characteristics

A total of 900 patients were included in the study: 413 (46%) from the internal COVID+ test set and 487 (54%) from the external COVID+ validation set (Fig 1, Table 1). The mean age of the internal cohort was 49.9 years (median = 51, range = 16–97, IQR = 39); 221 were White (53%), 31 were Asian (7%), 96 were Hispanic (23%), 27 were African American (6.5%), and 38 were other or unknown (10%). There were 4 deaths.

The mean age of the external validation set was 56.3 years (median = 57, range = 18–95, IQR = 45); 32 were White (6.6%), 225 were African American (46.2%), 59 were Hispanic (12.1%), and 165 were other (33.9%). In the external validation set, there were 59 deaths.

**Table 3. Precision and recall of Multi-Task Deep Learning HCC-Based Comorbidity and Sex Predictions for Internal and External Cohorts.**

| Characteristic | Internal validation | | External validation | |
|---|---|---|---|---|
| | Precision | Recall | Precision | Recall |
| Diabetes with chronic complications | 0.265 | 0.682 | 0.428 | 0.831 |
| Morbid obesity | 0.308 | 0.949 | 0.454 | 0.920 |
| Congestive Heart Failure | 0.079 | 0.909 | 0.382 | 0.625 |
| Cardiac Arrhythmias | 0.086 | 0.842 | 0.264 | 0.732 |
| Vascular Disease | 0.226 | 0.867 | 0.308 | 0.819 |
| COPD | 0.048 | 0.857 | 0.228 | 0.649 |
| All conditions | 0.166 | 0.800 | 0.364 | 0.752 |
| Sex | 0.844 | 0.928 | 0.884 | 0.942 |

Abbreviations: COPD = chronic obstructive pulmonary disease

**Table 4. Prognostic Model for Mortality in Hospitalized RT-PCR-Positive COVID-19 Patients from Deep Learning Predictors.**

| Variable | Adjusted odds ratio | 95% CI | *P* value |
|---|---|---|---|
| **Mortality Model** | | | |
| RAF | 34.96 | (10.22–119.54) | < .001 |
| COPD | 0.02 | (0.01–0.28) | .004 |
| Sex | 2.87 | (1.36–6.05) | .005 |
| Congestive heart failure | 0.06 | (0.01–0.56) | .014 |
| Diabetes with chronic complications | 0.07 | (0.01–0.59) | .014 |

Abbreviations: CI = confidence interval, COPD = chronic obstructive pulmonary disease, COVID-19 = coronavirus disease 2019, RAF = risk adjustment factor, RT-PCR = real-time reverse transcription polymerase chain reaction.

## Training cohort

A set of 11,257 anonymized unique frontal CXRs was used to train the CNN model previously described to predict patient sex, age, and six HCCs using administrative data. A randomly selected set of 2,864 (20%) radiographs was used as a first test set for initial model validation. The mean age at the time of the radiograph was 66 ± 13 years; 43% of the radiographs were from male patients.

## DL CNN analysis

The DL model produces a probability (0–1) for patient sex and each predicted HCC comorbidity. These predictions were compared to the HCC administrative data for the baseline, internal, and external validation cohorts. For each HCC, the relationship for the DL test data is summarized by a ROC and AUC (Table 2).

AUC predictions for the internal COVID+ cohort for each HCC ranged from 0.94 to 0.75 (Fig 2, Table 2 internal validation). The total AUC for all HCCs was 0.85. We then compared the HCC predictions on the external COVID+ validation set to administrative data to determine if the DL model was predictive. (Table 2, external validation). AUC ranged from 0.97 for sex to 0.63 for diabetes with chronic complications. The total AUC for all external validation cohort HCCs was 0.76. Comparison of the ROC AUC was performed between both cohorts and no statistically significant differences were found for COPD, morbid obesity, and cardiac arrhythmias (Table 2). Regarding recall and precision (Table 3), values followed the same trend as AUC, in the internal COVID+ cohort precision ranged from 0.05 to 0.84, and precision from 0.68 to 0.95. The external COVID+ cohort precision and recall values ranged from 0.23 to 0.89 and 0.63 to 0.94, respectively.

The correlation coefficient (R) between predicted and administrative RAF score in the internal validation set was 0.43 and in the external cohort was 0.36. The mean absolute error for the predicted RAF score in the internal validation set was 0.49 (SD ± 0.39) and in the external dataset was 1.2 (SD ± 1.04). In the administrative data, RAF scores were zero in 57.9% (n = 239) of the internal validation set patients (mean = 0.25, range = 0–4), and in 18.2% (n = 89) of the external patients (mean = 1.5, range 0–10).

Comparative and representative frontal CXRs from both internal and external COVID + cohort patients are shown in Fig 3, which demonstrate how the DL model analyzed the radiographs and generated the likelihoods of comorbidities. Qualitatively, the occlusion mapping demonstrated similar distributions in both even in the presence of artifacts such as image labels, electrodes, and rotated patient position. Additional paired t-testing of the internal validation set with augmented white text in the upper image corners did not generate a statistically

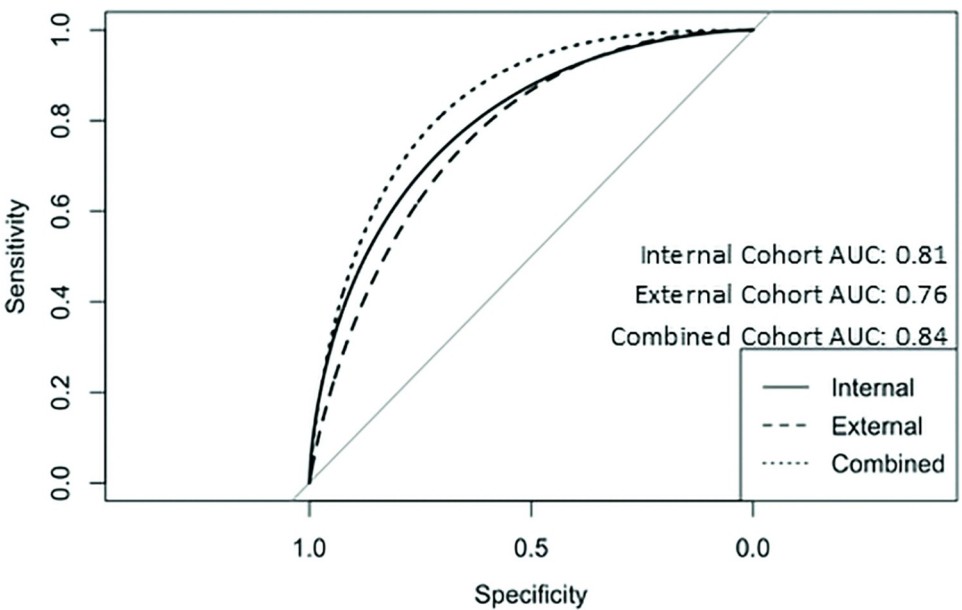

**Fig 2. Smoothed ROC curves and AUC ROC for predictive model applied to internal, external and combined cohorts.** Abbreviations: AUC = area under the curve, ROC = receiver operating characteristic.

significant difference at the $P$ = 0.05 level among all predictors. Additionally, paired t-testing was performed with input 8-bit PNG and 24-bit JPEG file formats, without a statistically significant difference at the $P$ = 0.05 level among all predictors [22].

Fig 4 describes the Spearman's rank correlation coefficients for each HCC and for RAF score for the EHR and the DL model for the entire dataset. Overall, comorbidities and the RAF score were positively correlated with each other, highlighting the comorbid nature of the chronic medical conditions evaluated, conditions with the highest correlation included being diagnosed with CHF, diabetes with chronic complications, vascular disease, heart arrhythmias and COPD. In addition, the RAF score from the DL model demonstrated good to strong correlation with five EHR covariables (0.52–0.9), consistent with the RAF score being a combination of HCCs.

## Univariate and multivariate analysis

The external validation set patients were significantly older (mean age, 56.3 years ± 16.4 vs 49.9 years ± 15, $P$ < .001) but similar in obesity (BMI 32.2 ± 10.1 vs 30.8 ± 7.08, $P$ = .01) compared with the internal validation set patients. From EHR data, the external validation set patients all had a significantly greater prevalence of comorbidities, as would be expected in a hospitalized cohort. The DL model predictions were correspondingly significantly larger in the external validation set for all six HCCs.

## Outcome modeling on external validation cohort

After backward elimination, the following variables remained in the logistic regression model for the prediction of mortality in the external validation cohort: RAF predicted score (adjusted OR, 35; 95% CI: 10–120; $P$ < .001), HCC11 (COPD) prediction (adjusted OR, 0.02; 95% CI: 0.001–0.29, $P$ = .004), male sex predicted (adjusted OR, 2.9, 95% CI: 1.36–6.05; $P$ < .01), HCC85 prediction (CHF) (adjusted OR, 0.058; 95% CI: 0.006–0.563; $P$ = .01), and HCC18

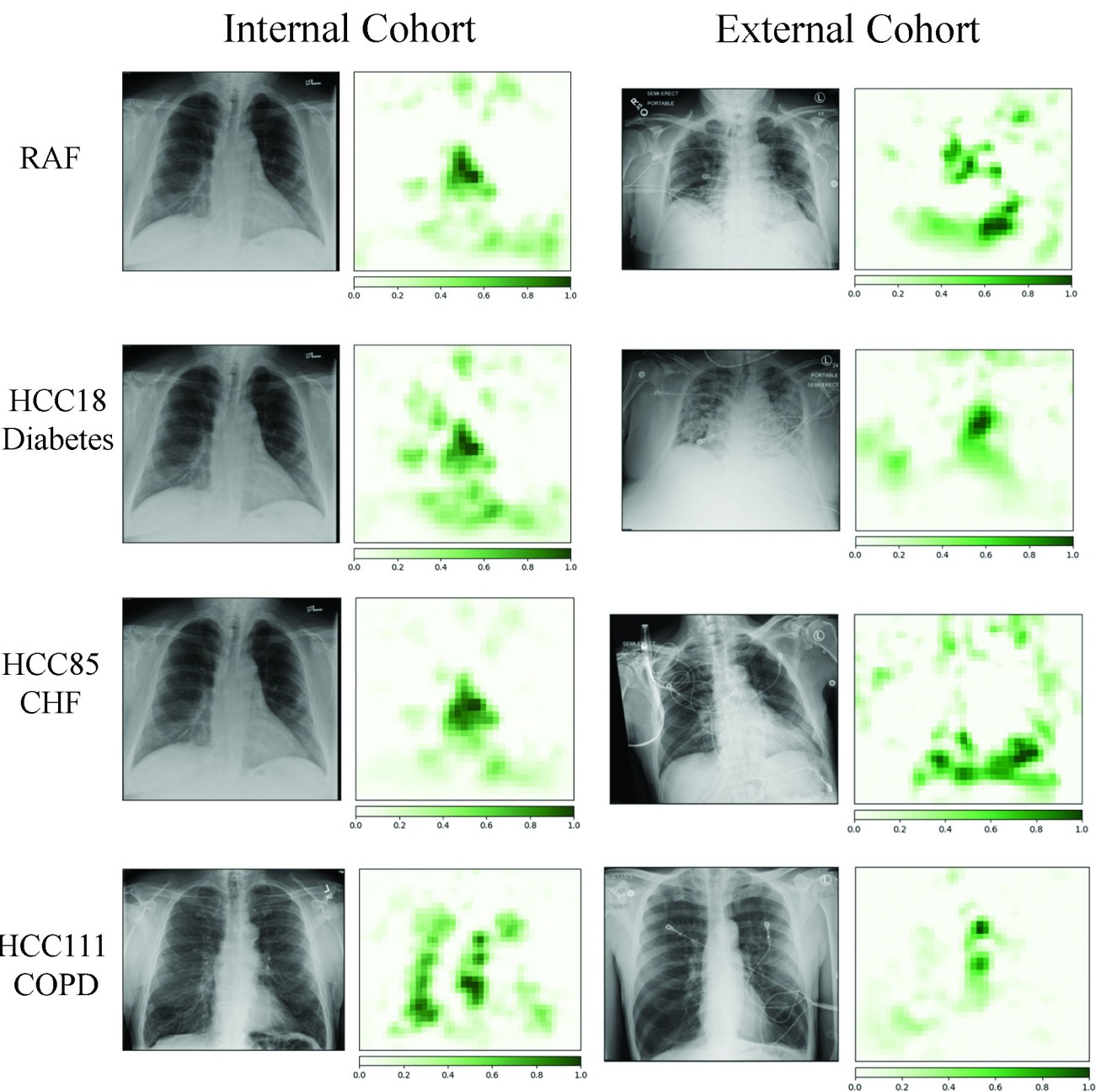

**Fig 3. Green pixels indicate a location that when occluded changes the risk adjustment factor (RAF) score in top row.** Each row represents the comorbidity and the resulting occlusion maps. Note that image labeling and electrodes do not impact the score. Please see discussion for further details.

prediction (diabetes with chronic complications) (adjusted OR, 0.07; 95% CI: 0.009–0.59; $P$ = .01) (Table 4). The AUC for this model was 0.76 (95% CI: 0.70–0.82). The model calibration had a slope of 0.92. Applying this logistic regression model on the combined internal validation and external validation cohorts (n = 900) demonstrated an ROC AUC of 0.84 (95% CI: 0.79–0.88) for the prediction of mortality, indicating the model has generalizability. The ROC AUC of the internal validation cohort modeled similarly was 0.81 with a nonsensical confidence interval due to the low number of mortalities in the ambulatory cohort of patients.

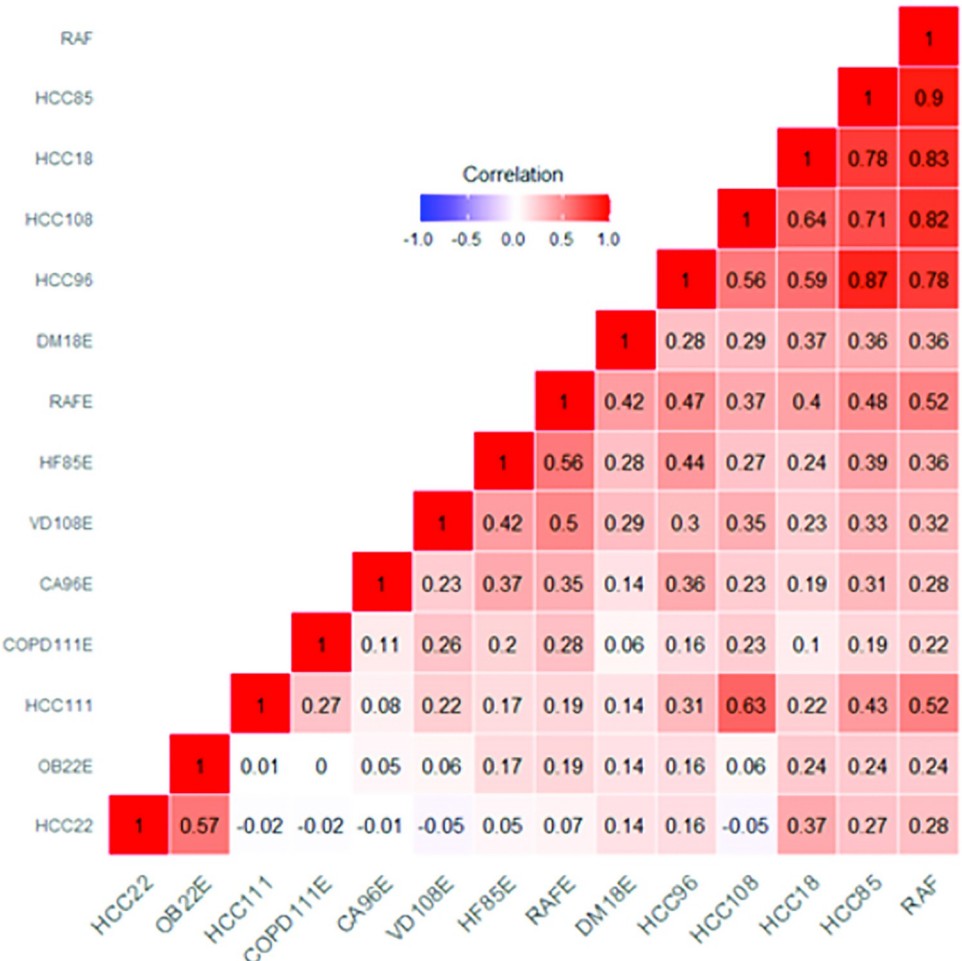

**Fig 4. Correlation heatmap of electronic health record (EHR) vs the deep learning (DL) model for both cohorts.**
The correlation matrix shows Spearman's correlation coefficient values across the predicted comorbidities and risk
adjustment factor (RAF) scores between the EHR data and the convolutional neural network (CNN) frontal
radiograph model. Please see the discussion for further details. Blue indicates a negative correlation, red indicates a
positive correlation. Abbreviations: DM18E = diabetes with complications, OBS22E = morbid obesity,
HF85E = congestive heart failure, CA96E = cardiac arrhythmias, CA96E = vascular disease, COPD111E = chronic
obstructive pulmonary disease, RAF = risk adjustment factor.

## Discussion

In this study we developed a DL model to predict select comorbidities and RAF score from
frontal CXRs and then externally validated this model using a VBC framework based on HCC
codes from ICD-10 administrative data. Because it was recognized that the internal validation
set was skewed towards less ill patients, with 4 deaths in the internal validation set, a corre-
sponding dataset with more severe disease was sought to ascertain validity over a broad spec-
trum of patient presentations. The DL model demonstrated discriminatory power in both
hospitalized external and ambulatory internal cohorts but was improved when operating on
the full spectrum of patients as would be seen across a region of practice with multiple different
care settings and degrees of patient illness. We feel justified in combining the cohorts which
together represent the full spectrum of COVID+ disease from ambulatory to moribund. The

covariables derived from the initial frontal CXRs of an external cohort of patients admitted with COVID-19 diagnoses were predictive of mortality.

The use of comorbidity indices derived from frontal CXRs has many potential benefits and can serve a dual role in VBC by identifying patients with an increased risk of mortality and therefore healthcare expenditures. Radiology's role in VBC has been frequently questioned [6], and imaging biomarkers, such as RAF score from CXRs, offer unique opportunities to identify undocumented, under-diagnosed, or undiagnosed high-risk patients. Increasingly, "full risk" VBC models require healthcare systems to bear financial responsibility for patients' negative outcomes, such as hospitalizations [23]. Radiologists currently may be unfamiliar with RAF scoring; however, US clinicians are increasingly aware because of the Medicare Advantage program and the associated reimbursement implications [24]. In our logistic regression model of the DL covariables, RAF score was a strong predictor of mortality; similarly, pre-existing administrative data is predictive of COVID-19 outcomes and hospitalization risk [4,25].

Many challenges exist within the fragmented US healthcare system, particularly in the utilization of medical administrative data. Since CXRs are frequently part of the initial assessment of many patients, including those with COVID-19, comorbidity scores predicted from CXRs could be rapidly available for and help clinicians with prognosis. We also found a significant percentage of patients in both cohorts who had RAF scores of 0 (excluding the demographic components), which could signify the absence of documented disease, or the absence of medical administrative data at the respective institution. However, there are conditions in the VBC HCC model that are difficult to confidently associate with a CXR, like major depression, and such a score would only be viewed as a part of a more comprehensive evaluation of the patient.

There are numerous clinical models of outcomes in COVID-19, many focused on admitted and critically ill hospitalized patients [26]. Several models have utilized the CXR as a predictor of mortality and morbidity for hospitalized COVID-19 patients, based on the severity, distribution, and extent of lung opacity [26]. We have previously demonstrated that features of the CXR other than those related to airspace disease can aid in prognostic prediction in COVID-19. In addition, a significant number of COVID-19 patients demonstrate subtle or no lung opacity on initial radiographic imaging, more likely early in the disease course and ambulatory setting, potentially limiting airspace-focused models [27].

Even before DL techniques, CXRs have been correlative for the risk of stroke, hypertension, and atherosclerosis through the identification of aortic calcification [28,29]. Similar DL methods were used on two large sets of frontal CXRs and demonstrated predictive power for mortality [30]. In another study [31], DL was used on a large public dataset to train a model to predict age from a frontal CXR, as age and comorbidities are correlated. These studies all suggest that the CXR can serve as a complex biomarker.

Our DL model allowed us to make predictions regarding the probabilities of comorbidities such as morbid obesity, diabetes, CHF, arrhythmias, vascular disease, and COPD; those were correlated with EHR diagnosis. Although these DL scores do not replace traditional diagnostic methods (i.e., HbA1c, BMI), they were predictive using the standard of EHR HCC codes. A heatmap of Spearman's correlation coefficient (Fig 4) between external and internal predictors from the DL model demonstrates that the RAF score is an amalgamation of the other comorbidities scores, and is strongly correlated to cardiovascular predictors, such as CHF (HCC85) and vascular disease (HCC108).

The DL model's performance was diminished in the external cohort, as seen by lower AUCs in several prediction classes. Reasons for this may include differences in disease prevalence in the populations, institutional differences in documenting disease, external artifacts, and potential differences based on race, ethnicity, socioeconomic factors. One clear difference between the training and internal validation cohort as compared to the external validation

cohort was that the CNN was trained on PA CXRs, and the internal cohort used PA radiographs, but the external cohort had less than 10% PA CXRs. It is likely that the CNN would have performed better on the external COVID+ patients if the CNN had been trained on similar patients using PA CXRs. However, despite all these differences, the DL model maintained discriminatory ability.

Occlusion maps (Fig 3) did not demonstrate significant attribution to external artifacts, such as image markers, with overall similar patterns of attribution between the two cohorts per class. The occlusion mapping in our cohorts demonstrates positive attribution to the cardiovascular structures for the RAF score, such as the heart and aorta, which was a strong predictor for mortality in the external cohort. Whereas in the prediction for COPD (HCC111) the hyperinflated lung parenchyma and a narrow mediastinum are predominant features on occlusion mapping (Fig 3), similar to what a radiologist would observe. The prediction of diabetes utilizes body habitus at the upper abdomen, axillary regions, and lower neck (Fig 3), which is strongly associated with type II diabetes [32]. Using logistic regression on the DL model's predictions in the external cohort against mortality demonstrated ROC AUC values that were comparable to other studies [11,33]. This logistic regression model, when applied to both cohorts demonstrated an increased ROC AUC, as the internal cohort had less disease findings, similarly described by Schalekamp [11].

Our study was limited by several factors, in the internal validation cohort, by incomplete hospitalization records and laboratory assessments, and small endpoint analysis. Within the ambulatory setting, many patients had imaging at other locations, which were not available for comparison in this study, as well as a certain self-selection bias of presenting directly to the hospital when seriously ill. In the external cohort, the extensive use of portable CXRs, patient positioning, and artifacts (electrodes, labeling) may have impacted the results because of increased noise. Lastly, implementation of DL models remains a technical challenge for many institutions and practices, with relatively few standard platforms or widespread consistent adoption.

In conclusion, we found that a multi-task CNN DL model of comorbidities derived from VBC-based administrative data was able to predict 6 comorbidities and mortality in an ambulatory COVID+ cohort as well as an inpatient COVID+ cohort. In addition, these model covariables, especially the VBC-based predicted RAF, could be used in logistic regression to predict mortality without any additional laboratory or clinical measures. This result suggests further validation and extension of this particular methodology for potential use as a clinical decision support tool to produce a prognosis at the time of the initial CXR in a COVID + patient, and perhaps more generally as well.

## Supporting information

**S1 File. Detecting Racial/Ethnic Health Disparities Using Deep Learning From Frontal Chest Radiography.** Pyrros A, Rodríguez-Fernández JM, Borstelmann SM, Gichoya JW, Horowitz JM, Fornelli B, Siddiqui N, Velichko Y, Koyejo Sanmi O, Galanter W. Detecting Racial/Ethnic Health Disparities Using Deep Learning From Frontal Chest Radiography. J Am Coll Radiol. 2022 Jan;19(1 Pt B):184–191. doi: 10.1016/j.jacr.2021.09.010. Erratum in: J Am Coll Radiol. 2022 Mar;19(3):403. PMID: 35033309; PMCID: PMC8820271. (PDF)

**S2 File. Predicting Prolonged Hospitalization and Supplemental Oxygenation in Patients with COVID-19 Infection from Ambulatory Chest Radiographs using Deep Learning.** Pyrros A, Flanders AE, Rodríguez-Fernández JM, Chen A, Cole P, Wenzke D, Hart E, Harford S, Horowitz J, Nikolaidis P, Muzaffar N, Boddipalli V, Nebhrajani J, Siddiqui N, Willis M, Darabi

H, Koyejo O, Galanter W. Predicting Prolonged Hospitalization and Supplemental Oxygenation in Patients with COVID-19 Infection from Ambulatory Chest Radiographs using Deep Learning. Acad Radiol. 2021 Aug;28(8):1151–1158. doi: 10.1016/j.acra.2021.05.002. Epub 2021 May 21. PMID: 34134940; PMCID: PMC8139280.
(PDF)

**S1 Table. A representative example demonstrating how the risk adjustment factor (RAF) is calculated for a female patient aged 65–69 years.** Sample ICD10 hierarchical condition category (HCC) codes map to their respective HCC categories. In this example, the patient has codes for both diabetes with chronic complication and without complications; the hierarchy means HCC18 supersedes HCC19, and the lower coefficient is dropped. In addition, there is a diagnosis for congestive heart failure (CHF), which results in a disease interaction raising the RAF score by an additional coefficient. Lastly, having more than one ICD10 code per category does not alter the coefficients. *In our deep learning model, we excluded the demographic component to separately control for age and sex.
(DOCX)

**S1 Data. Deidentified data of 900 patients.**
(CSV)

## Author Contributions

**Conceptualization:** Ayis Pyrros, Jorge Rodriguez Fernandez, William Galanter.

**Data curation:** Ayis Pyrros, Jorge Rodriguez Fernandez, Melinda Willis, Patrick Cole.

**Formal analysis:** Ayis Pyrros, Jorge Rodriguez Fernandez, Stephen M. Borstelmann, Patrick Cole, William Galanter.

**Investigation:** Ayis Pyrros, Jorge Rodriguez Fernandez, Nadir Muzaffar, William Galanter.

**Methodology:** Jorge Rodriguez Fernandez, Andrew Chen, Sanmi Koyejo, William Galanter.

**Project administration:** Ayis Pyrros, Jorge Rodriguez Fernandez, Sanmi Koyejo, William Galanter.

**Resources:** Jorge Rodriguez Fernandez, Patrick Cole, Sanmi Koyejo.

**Software:** Ayis Pyrros, Jorge Rodriguez Fernandez, Patrick Cole, Sanmi Koyejo.

**Supervision:** Ayis Pyrros, Jorge Rodriguez Fernandez, Adam Flanders, Daniel Wenzke, Paul Nikolaidis, Melinda Willis, Andrew Chen, Patrick Cole, Nasir Siddiqui, Sanmi Koyejo, William Galanter.

**Validation:** Ayis Pyrros, Andrew Chen, Jennifer McVean, William Galanter.

**Visualization:** Stephen M. Borstelmann, Andrew Chen, Nasir Siddiqui, Sanmi Koyejo, William Galanter.

**Writing – original draft:** Ayis Pyrros, Jorge Rodriguez Fernandez, Stephen M. Borstelmann, Daniel Wenzke, Eric Hart, Jeanne M. Horowitz, Paul Nikolaidis, Melinda Willis, Andrew Chen, Patrick Cole, Nasir Siddiqui, Momin Muzaffar, Jennifer McVean, Martha Menchaca, Aggelos K. Katsaggelos, Sanmi Koyejo, William Galanter.

**Writing – review & editing:** Ayis Pyrros, Jorge Rodriguez Fernandez, Stephen M. Borstelmann, Adam Flanders, Daniel Wenzke, Eric Hart, Jeanne M. Horowitz, Paul Nikolaidis,

Melinda Willis, Andrew Chen, Nasir Siddiqui, Momin Muzaffar, Jennifer McVean, Martha Menchaca, Aggelos K. Katsaggelos, Sanmi Koyejo, William Galanter.

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
