## [Decision Letter · Decision Letter 0]

25 Mar 2022

PDIG-D-22-00013

Validation of a Deep Learning, Value-Based Healthcare Model to Predict Mortality and Comorbidities from Chest Radiographs in COVID-19

PLOS Digital Health

Dear Dr. Pyrros,

Thank you for submitting your manuscript to PLOS Digital Health. After careful consideration, we feel that it has merit but does not fully meet PLOS Digital Health's publication criteria as it currently stands. Therefore, we invite you to submit a revised version of the manuscript that addresses the points raised during the review process.

Please address all of the reviewer comments and suggestions, especially in the Methods section. More details and analyses are needed before the paper can be published. 

We look forward to receiving your revised manuscript.

Kind regards,

Heather Mattie

Academic Editor

PLOS Digital Health

Journal Requirements:

1. Please amend your detailed Financial Disclosure statement. This is published with the article, therefore should be completed in full sentences and contain the exact wording you wish to be published.

i). State the initials, alongside each funding source, of each author to receive each grant.

ii). State what role the funders took in the study. If the funders had no role in your study, please state: “The funders had no role in study design, data collection and analysis, decision to publish, or preparation of the manuscript.”

2. Please update your Competing Interests statement. If you have no competing interests to declare, please state: “The authors have declared that no competing interests exist.”

3. Please note that your Data Availability Statement is currently missing the direct link to access each database. If your manuscript is accepted for publication, you will be asked to provide these details on a very short timeline. We therefore suggest that you provide this information now, though we will not hold up the peer review process if you are unable.

4. Please ensure that the Title in your manuscript file and the Title provided in your online submission form are the same.

5. Please include the title page at the beginning of your main manuscript and not as a separate file.

6. Please change the file type of 1-s2.0-S1076633221002178-main.pdf and JACR-Article-main.pdf to “Supporting Information” and ensure that each one has a legend listed in the manuscript after the references list.

7. Please provide separate figure files in .tif or .eps format only and remove any figures embedded in your manuscript file. Please ensure that all files are under our size limit of 20MB.

8. We notice that your supplementary table (Table SI-1) is included in the manuscript file. Please remove them and upload them with the file type 'Supporting Information'. Please ensure that all Supporting Information files are included correctly and that each one has a legend listed in the manuscript after the references list.

Additional Editor Comments (if provided):

Reviewers' comments:

Reviewer's Responses to Questions

**Comments to the Author**

1. Does this manuscript meet PLOS Digital Health’s publication criteria? Is the manuscript technically sound, and do the data support the conclusions? The manuscript must describe methodologically and ethically rigorous research with conclusions that are appropriately drawn based on the data presented.

Reviewer #1: Yes

Reviewer #2: Yes

2. Has the statistical analysis been performed appropriately and rigorously?

Reviewer #1: Yes

Reviewer #2: No

3. Have the authors made all data underlying the findings in their manuscript fully available (please refer to the Data Availability Statement at the start of the manuscript PDF file)?

Reviewer #1: No

Reviewer #2: Yes

4. Is the manuscript presented in an intelligible fashion and written in standard English?

Reviewer #1: Yes

Reviewer #2: Yes

5. Review Comments to the Author

Reviewer #1: This paper aims to validate the deep learning model for mortality and comorbidity prediction in COVID-19 patients. The paper was well-written. The ideal is novel and interesting but the descriptions of methodology and the whole pipeline are not clear enough. After reading the methods section (especially the modeling part), I still don’t really understand what exactly this work has done. Thus, I am not able to evaluate the contribution of this study with the current presentation. Some comments are given below: 

Major:

1.The methods were not detailed enough for others to reproduce their works. I don’t understand the role of feature selection. What’s the task by which the AUC was calculated? I am worried about the data leakage problem occuring in this step. Did they use a standalone data split for feature selection? They also mentioned the use of multivariable logistic regression and backward elimination with fractional polynomial transformation to include the variables from the DL. What is the purpose and process of backward elimination? What are the results if they don’t go through this step?

2.What is the task when pretraining the CXR model on CheXpert? 

3.Did authors split the dataset into training, validation and testing and use validation set for hyperparameters selection (or feature selection)? 

4.The authors mentioned that the MTL can exploit the inter-task relationships but I didn’t see such relationships in their results. I am wondering whether there is an improvement from separate learning? I think they need some implementation details but not only general introduction in Method section and some comparison results in Results section to prove that the MTL is helpful. 

5.In addition to AUC, the authors should report the results with the other evaluation metrics (e.g., recall and precision) since the patient numbers of each medical condition and mortality are very small. 

6.I would suggest that the authors give more explanation on the visualization maps to see if the model is looking at the meaningful locations.

7.Similarly, the authors could give more explanation on Figure 3. There are some other high correlation values besides those five variables.

Minor:

1.In the first part of the Results section. Why was the mean age described here? “The mean age at the time of the radiograph was 66 ± 13.263 years”. I think it should belong to the data description section.

2.Do horizontal flips of the CXR images make sense? Especially when the model is trying to predict congestive heart failure. 

3.Add information in Figure 4 (or its caption) to directly let the reader know it is for mortality prediction.

Reviewer #2: In this manuscript, the authors report development and validation of a DL model to predict comorbidities and mortality from chest X rays in COVID-19 patients. The authors have described their model, data and interpretations in details and, in general, the manuscript is well written. I have a few suggestions that can further improve the manuscript.

1. In the Introduction section the authors make a very logical case for using a DL model to predict comrobidities (and some sociodemographics) from chest X-rays. However, the reasoning to convert this into a mortality predictor is somewhat assumed as a natural consequence of the comorbidity prediction. Consequently, in the rest of the manuscript the authors use a logistic regression model for prognostication. The authors claim that this prognostic model is generalizable since it gave an AUC of 0.84 in the combined (internal and external cohort). I have several concerns with regard to this claim:

a. I am concerned that the value of AUC reported in the combined cohort is higher than the individual components (0.81 in internal and 0.76 in external). Please check the numbers.

b. The model seems to include repeated information (Table 3). For example, comorbidities like COPD, CHF and Diabetes have ORs that indicate a protective role against mortality. I think this is happening because the RAF score already includes information on comorbidities and thus including each comorbidity again in the full model isn't the way to approach this. 

c. I think that the prognostic model needs to be re-specified. Why not use all elements of the RAF score and the predicted sociodemographics to run a machine learning model (RF, SVM or XGBoost) to predict mortality? Also do this separately for the internal and external cohorts (please don't combine them).

2. The authors have modified the ResNet34 model to improve the classification performance for prediction of the comorbidities. However, in the entire manuscript there is no direct comparison of the performance of the standard ResNet34 with the improved model that the authors used. I suggest that the model accuracy of ResNet34 be compared against that of this improved model.

3. Extending this line of thinking, the authors also need to compare the performance of other CNN architectures – deeper ResNets (101, 152), DenseNet (at least 121), EfficientNets (at least B1), Inception (v3) and Xception. It is possible that the performance of some of these candidate alternatives may be better than the architecture used by the authors. If not, then all the reason to use the authors’ model!

4. Numbers reported in Table 2 indicate that there was substantial overfitting for some conditions (CHF, COPD, Vascular disease and somewhat for diabetes). The authors need to describe whether this potential overfitting was despite measures like regularization and dropouts? If not, the authors need to add these (and other) measures to reduce overfitting and thus improve the classification performance in the external dataset.

5. The manuscript needs to be copyedited for typos.

6. PLOS authors have the option to publish the peer review history of their article (what does this mean?). If published, this will include your full peer review and any attached files.

**Do you want your identity to be public for this peer review?** For information about this choice, including consent withdrawal, please see our Privacy Policy.

Reviewer #1: No

Reviewer #2: No

---

## [Editor Report · Decision Letter 1]

5 May 2022

Validation of a deep learning, value-based care model to predict mortality and comorbidities from chest radiographs in COVID-19

PDIG-D-22-00013R1

Dear Dr Pyrros,

We are pleased to inform you that your manuscript 'Validation of a deep learning, value-based care model to predict mortality and comorbidities from chest radiographs in COVID-19' has been provisionally accepted for publication in PLOS Digital Health.

Best regards,

Heather Mattie

Academic Editor

PLOS Digital Health